evolution, behaviour, ecology

social learning, foraging, competition, phenotypic divergence, adaptive dynamics, disruptive selection

**Author for correspondence:**
R. Tucker Gilman
e-mail: tucker.gilman@manchester.ac.uk

# Competition for resources can promote the divergence of social learning phenotypes

R. Tucker Gilman[1], Fern Johnson[2] and Marco Smolla[3]

[1]Department of Earth and Environmental Sciences and [2]Faculty of Biology, Medicine and Health, University of Manchester, Manchester UK
[3]Department of Biology, University of Pennsylvania, Philadelphia, PA USA

RTG, 0000-0003-2000-7966; MS, 0000-0001-6367-8765

Social learning occurs when animals acquire knowledge or skills by observing or interacting with others and is the fundamental building block of culture. Within populations, some individuals use social learning more frequently than others, but why social learning phenotypes differ among individuals is poorly understood. We modelled the evolution of social learning frequency in a system where foragers compete for resources, and there are many different foraging options to learn about. Social learning phenotypes diverged when some options offered much better rewards than others and expected rewards changed moderately quickly over time. When options offered similar rewards or when rewards changed slowly, a single social learning phenotype evolved. This held for fixed and simple conditional social learning rules. Sufficiently complex conditional social learning rules prevented the divergence of social learning phenotypes under all conditions. Our results explain how competition can promote the divergence of social learning phenotypes.

## 1. Introduction

Many animals acquire skills or knowledge by observing or interacting with others [1]. This social learning is in contrast with individual learning, in which animals learn by first-hand experience or trial-and-error. Social learning is widespread in nature. It has been observed in mammals, birds, fishes [2], and social and non-social invertebrates [3–6]. Social learning can impact ecological interactions [7–9], habitat use [10], and micro- and macroevolutionary processes [11,12] and has garnered attention for its role as the fundamental building block of culture [1,4,13].

Rates of social learning often differ among individuals in the same population, so that some individuals use social learning more frequently than others [14,15]. In some cases, differences in social learning phenotypes can be linked to differences in life history [16]. For example, in guppies, foraging information spreads faster in the groups of females than in the groups of males [17]. According to Reader & Laland [17], females may be more likely to use social learning when foraging because their fitness depends on reliable resource acquisition, while male fitness depends more immediately on access to females. Social learning rates may also differ if individuals have traits that confer different aptitudes for the learning types, so that individuals with some traits are intrinsically better at social learning and individuals with other traits are intrinsically better at individual learning [18]. Moreover, social learning rates can differ if individuals have different opportunities for learning. For example, in Japanese macaques, females and juvenile males forage in groups and acquire socially transmitted foraging behaviours quickly, but adolescent and adult males often forage alone and acquire socially transmitted foraging behaviours

slowly or not at all [19]. However, in some populations, social learning phenotypes differ even when other differences among individuals are not apparent. For example, Aplin & Morand-Ferron found differences in social learning frequency among great tits, even without clear pre-existing differences among birds [20]. Similarly, chimpanzees have different social learning frequencies within sexes, and the adaptive value of these differences remains unexplained [15].

Understanding how and why social learning phenotypes differ among individuals is important for at least two reasons. First, how the use of social learning varies among members of a population affects how information is transmitted and maintained [17], and so may affect the emergence and accumulation of culture [4,14,21]. Second, Arbilly et al. [22] found that differences in social learning phenotypes can lead to the divergence of other traits or behaviours. If differences in social learning promote the divergence of clusters of behavioural traits, and if these trait clusters are expressed consistently across ecological contexts, then the divergence of social learning phenotypes may help to explain the origin of animal personality.

Although differences in social learning frequencies have been widely observed in nature, theory to explain how social learning phenotypes diverge in otherwise homogeneous populations is lacking [20,23]. Much social learning theory originates with classical producer-scrounger models (e.g. [24,25–28]), in which producers are individual learners and scroungers are social learners. These models predict the proportion of social learning that will evolve in a given population, but do not predict the divergence of social learning phenotypes within populations [23]. For example, Dubois et al. [26] found that two suboptimal social learning phenotypes could persist in a population, but if a single phenotype with an optimal proportion of social learning appeared, then that phenotype would exclude all others. More recently, Smolla et al. [29] studied a model in which individuals forage for resources in a large system of patches that varies in resource quality in both time and space. Foragers could find resources by searching randomly or by copying others. Social learning in the population did not converge to a single phenotype. However, Smolla et al.'s model was stochastic and included a high rate of mutation to the social learning phenotype, so the authors could not determine whether variability in the social learning phenotype was adaptive or was maintained by mutation and drift. Thus, how and under what conditions variability in social learning phenotypes is maintained remains an open question.

In this study, we constructed a deterministic version of Smolla et al.'s [29] model and analysed it using tools from adaptive dynamics [30,31]. Our analysis does not rely on a high rate of mutation to maintain genetic variability and enable evolution. We identified the conditions under which foraging competition, a ubiquitous attribute of ecological systems, selects for the divergence of social learning phenotypes within populations. Then, we examined cases in which conditional social learning rules might resolve disruptive selection and prevent the divergence of social learning phenotypes.

## 2. Basic model

We modelled an infinite population foraging in continuous time for resources distributed over an infinite number of patches without spatial structure. We describe patches as foraging sites and social learning as site choice copying, but our model would also apply if patches represent foraging techniques (e.g. nut-hammering, termite-fishing or pestle-pounding by chimpanzees, [32]) and social learning represents the copying of those techniques. Patches can be either 'good' or 'bad'. Good patches offer resource value $r$, and bad patches offer no resources. Patches experience change events at rate $c$. A patch that experiences a change event becomes (or remains) good with probability $g$ and becomes (or remains) bad with probability $1 - g$. Thus, the proportion of good patches in the system is $g$, and the unevenness of the resource distribution is described by $1 - g$ (i.e. the Gini coefficient [33]).

At any point in time, each forager in our model occupies a patch. If a forager occupies a good patch, it collects resources at rate $r/n^q$, where $n$ is the number of foragers in the patch and $q$ controls the strength of competition. We assumed that resources are replenished after they are consumed (e.g. flowers that replenish their nectar [34]), so $r$ remains constant while the patch remains good. If a forager occupies a bad patch, it collects no resources.

Foragers experience learning events at rate $l$. A learning event can be social (with probability $s$) or individual (with probability $1 - s$). If a forager learns socially, it randomly selects another forager from the population, and it learns the rate at which it would collect resources if it joined that forager in its patch. If a forager learns individually, it randomly selects a patch in the system and learns the rate at which it would collect resources if it moved to that patch. If a forager learns about a patch where it would collect more resources than in its current patch, it moves instantly to that patch with no cost of moving. Otherwise, it remains in its current patch. Foragers do not retain information about patches. Thus, at the time of learning, a forager knows the resource collection rate in only two patches: its current patch and the patch it is learning about. We set $l = 1$ for all simulations. This scales time in the model to the learning rate but does not affect the generality of the model.

For computational convenience, we assumed that patches with $n_{max}$ foragers are 'full'. A forager that learns about such a patch cannot move to that patch and remains in its current patch. This assumption has little effect on how foragers are distributed among patches. At the ideal free distribution, the number of foragers in each good patch would be $D/g$, where $D$ is the density of foragers per patch in the system. Although the ideal free distribution is not reached in systems where patch values change, patches with many more than $D/g$ foragers are rare in our model. We set $n_{max}$ so that no more than 1 in $10^6$ patches have $n_{max}$ foragers, and we did not investigate the role of $n_{max}$ further in this study.

## 3. Analysis

### (a) Overview

Our goal was to understand whether the proportion of social learning, $s$, in the systems we modelled evolves to an optimum shared by all members of the population or diverges within populations. We approached this question using the adaptive dynamics framework [31]. We began by modelling a population that is monomorphic for a social learning phenotype $s_r$ and is at its steady state distribution across good and bad patches. We introduced a rare mutant with a slightly different social learning phenotype, $s_m$, to the population, and

found the steady state distribution of the mutant in the system dominated by the resident. We calculated the fitness of the resident and mutant at their steady state distributions. If the fitness of the mutant was higher than that of the resident, then the mutant can invade and replace the resident in the population [30]. Thus, we replaced the resident with the mutant, and we introduced a new mutant phenotype into the population. We iterated this process and tracked the evolution of the social learning phenotype until it reached either (i) a phenotype where no further mutations could invade (i.e. an evolutionarily stable state), or (ii) a phenotype where mutations that increase or decrease the social learning phenotype could both invade (i.e. an evolutionary branching point). Because adaptive dynamics studies the invasion of mutations into monomorphic populations, we cannot track the evolution of social learning phenotypes after they diverge. However, the existence of an evolutionary branching point implies that evolution favours the diversification of social learning phenotypes [30,31].

### (b) Finding the steady state distribution of the resident population

Patches in our model are characterized by their quality (i.e. good or bad) and their occupancy. Thus, the state of the system can be characterized by a probability vector $\mathbf{v}$ of length $2 + 2 n_{max}$, where entry $v_i$ is the proportion of patches that are good and have $i - 1$ occupants for $i \in \{1, 2, \dots, 1 + n_{max}\}$, and is the proportion of patches that are bad and have $i - (2 + n_{max})$ occupants for $i \in \{2 + n_{max}, 3 + n_{max}, \dots, 2 + 2 n_{max}\}$. We tracked the state of the system through time using a system of ordinary differential equations:

$$\frac{d\mathbf{v}}{dt} = \mathbf{M}(\mathbf{v})\mathbf{v}, \tag{3.1}$$

where entry $m_{ij}(\mathbf{v})$ of matrix $\mathbf{M}(\mathbf{v})$ is the rate at which patches in state $j$ transition to state $i$, and depends on the current state $\mathbf{v}$ of the system. We computed $\mathbf{M}(\mathbf{v})$ (electronic supplementary material, S1) and solved

$$0 = \mathbf{M}(\mathbf{v}^*)\mathbf{v}^*, \tag{3.2}$$

to find the steady state distribution $\mathbf{v}^*$ for each parameter set that we wished to study.

### (c) Finding the steady state distribution of the rare mutant

We introduced a mutant with a social learning phenotype $s_m$ to the resident population in its steady state. We assumed that the mutant is rare enough that (i) no mutant ever encounters another mutant, and (ii) the mutant does not affect the state of the resident population. Each mutant is characterized by the quality of the patch it occupies and the number of residents that share its patch. Thus, the state of the mutant population can be characterized by a probability vector $\mathbf{u}$ of length $2 + 2 n_{max}$, where entry $u_i$ is the proportion of mutants that are in good patches shared by $i - 1$ residents for $i \in \{1, 2, \dots, 1 + n_{max}\}$ and is the proportion of mutants that are in bad patches shared by $i - (2 + n_{max})$ residents for $i \in \{2 + n_{max}, 3 + n_{max}, \dots, 2 + 2 n_{max}\}$. A mutant cannot share a patch with $n_{max}$ residents, so $u_{1+n_{max}} = u_{2+2 n_{max}} = 0$. We included these entries in $\mathbf{u}$ so that each entry of $\mathbf{u}$ has the same number of residents as the corresponding entry of $\mathbf{v}$,

which facilitates analysis later. We tracked the state of the mutant population through time using

$$\frac{d\mathbf{u}}{dt} = \mathbf{N}\mathbf{u}, \tag{3.3}$$

where entry $n_{ij}$ of matrix $\mathbf{N}$ is the rate at which mutants in state $j$ transition to state $i$. Because the mutant is rare, the entries of $\mathbf{N}$ depend on $\mathbf{v}^*$ but not on $\mathbf{u}$ (electronic supplementary material, S2). Thus, the steady state distribution of the mutant population, $\mathbf{u}^*$, is the solution to

$$0 = \mathbf{N}\mathbf{u}^*. \tag{3.4}$$

### (d) Comparing the fitness of the resident and mutant phenotypes

Following classical foraging theory, we assumed that each forager's fitness is proportional to the resources it collects [35]. Thus, the mean fitness of the resident phenotype is

$$w_{res} = \frac{1}{D} \sum_{i=2}^{1+n_{max}} \frac{(i - 1)}{(i - 1)^q} v_i^* r, \tag{3.5}$$

and the mean fitness of the mutant phenotype is

$$w_{mut} = \sum_{i=1}^{n_{max}} \frac{u_i^*}{i^q} r. \tag{3.6}$$

If $w_{mut} > w_{res}$, we assumed that the mutant would invade and replace the resident, and otherwise the resident would repel the invasion. This assumption is valid for small mutations to the resident phenotype [30]. The value of $r$ does not affect the inequality $w_{mut} > w_{res}$, and so does not affect the behaviour of our model. Therefore, we simplify the model by setting $r = 1$.

### (e) Finding critical points and testing for stability or evolutionary branching

For each set of parameter values that we studied, we initialized the model with a resident population that used only individual learning (i.e. $s_r = 0$), and we tested the resident for invasion by a mutant with $s_m = 0.001$. If the mutant invaded, we replaced the resident with the mutant (i.e. we set $s_r = 0.001$) and we tested for invasion by new mutants with $s_m = s_r + 0.001$ and $s_m = s_r - 0.001$. We continued this process as long as mutants with higher social learning could invade and mutants with lower social learning could not. If we reached a point where the mutant with higher social learning could not invade but the mutant with lower social learning could, then we took two steps back (i.e. to $s - 0.002$), we reduced the step size by a factor of 10 (i.e. from 0.001 to 0.0001), and we continued the analysis. This allowed us to move progressively closer to a convergence-stable value of $s$. We iterated this process until we reached a value of $s_r$ that (i) could not be invaded by either mutant, or (ii) could be invaded by both mutants. Case (i) represents a locally stable state and indicates the social learning phenotype we should expect to evolve under that parameter combination. Case (ii) represents an evolutionary branching point. In this case, we should expect the social learning phenotype to evolve to this point and then diverge in the population. If the system reached a locally stable state, we tested that phenotype for invasion by all social learning phenotypes in the

set $s_m \in \{0, 0.01, 0.02, \ldots, 1\}$ to discover whether the locally stable phenotype was also globally stable. To help visualize the evolutionary dynamics in the systems we modelled, we created pairwise invasibility plots (i.e. PIPs, [31]) for a subset of the parameter combinations. PIPs show whether each possible resident phenotype can be invaded by a range of mutant phenotypes and offer graphical representations of evolutionary dynamics.

## (f) Studying the effects of model parameters

Our model has four parameters: the rate of environmental change, $c$; the environmental heterogeneity, controlled by $g$; the strength of competition, $q$; and the population density $D$. Our analysis focused on the effects of $c$ and $g$. We fixed $q = D = 1$ and conducted deterministic simulations to ask whether divergence of the social learning phenotype is favoured for combinations of $c \in [0.15, 0.80]$ and $g \in [0.05, 0.45]$. In the electronic supplementary material, figure S1, we show that our qualitative results are not limited to the special case in which $q = D = 1$.

Matlab codes for models presented in this paper are vailable from the Dryad repository at 10.5061/dryad. v6wwpzgrm [36].

## 4. Results

We tested our model by comparing our results to those of Smolla *et al.* [29]. Like Smolla *et al.*'s model, our model predicts that social learning will evolve when resource distributions are uneven and when the rate at which patch quality changes is not too fast (figure 1).

Social learning can evolve to an evolutionarily stable state or to an evolutionary branching point, depending on the resource distribution and the rate at which patch quality changes (figure 1). When patch quality changes slowly, social learning evolves to an evolutionarily stable state in which all foragers have the same probability of social learning. When resources are unevenly distributed among patches and patch quality changes quickly, the probability of social learning evolves to an evolutionary branching point. From this point, selection favours the diversification of social learning phenotypes among foragers.

The divergence of social learning phenotypes is favoured if a forager that has recently used one learning type can expect to benefit using the same learning type again (electronic supplementary material, S3). In our model, this happens because the different learning types tend to discover different patches. Individual learning is more likely to discover a bad patch, but is also more likely to discover a good patch with few occupants. A forager that finds a good patch with few occupants will move to that patch. From there, it can increase its fitness only if it finds another good patch with even fewer occupants, and individual learning is more likely than social learning to discover such a patch. Thus, after a forager has discovered a good patch with few occupants by individual learning, it is advantageous for that forager to use individual learning again (electronic supplementary material, figure S2).

Evolutionary branching of social learning phenotypes occurs only when the resource distribution is uneven and the rate of change is relatively fast (figure 1). If the resource distribution is even, then individual learning usually discovers good patches, and social learning does not evolve. If the rate of

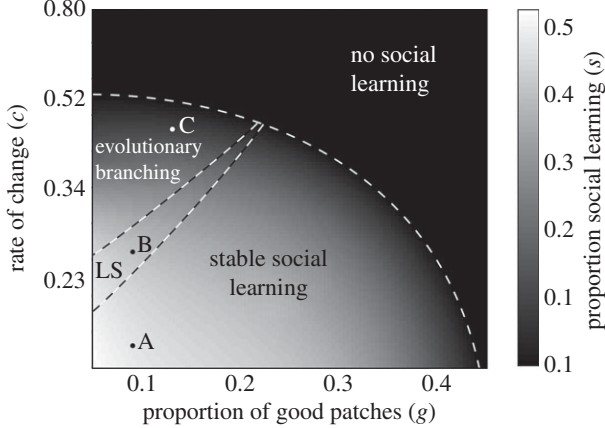

**Figure 1.** The proportion and evolutionary stability of social learning that evolves under different combinations of environmental heterogeneity and environmental change. Environmental heterogeneity decreases as $g$ increases, and the rate of environmental change increases as $c$ increases. Social learning evolves when environments are heterogeneous and change is slow. If the environment is sufficiently heterogeneous and change is not too slow, then social learning evolves to an evolutionary branching point. In such cases, we expect the population to evolve multiple social learning phenotypes. In a small part of parameter space (LS), the evolved social learning phenotype is locally stable. In such cases, social learning phenotypes can diverge only if mutations to the phenotype have large effects. A, B and C show points corresponding to the pairwise invasibility plots in figure 2.

change is slow but the resource distribution is uneven, then the system reaches a steady state where most foragers are in densely occupied good patches. Because these patches are densely occupied, they are most likely to be encountered by social learning. However, for foragers in these patches, individual learning is advantageous because it is more likely than social learning to discover lower occupancy patches that offer higher resource collection rates. At the same time, foragers that have just learned individually are more likely to occupy bad patches, because social learning usually finds good albeit densely occupied patches. However, for foragers in bad patches, social learning is advantageous, because it discovers patches where foragers can start collecting at least some resources quickly. Because each learning type tends to lead foragers to patches where the other learning type is advantageous, the optimal strategy is for each forager to mix learning types, and social learning phenotypes do not diverge (electronic supplementary material, figure S3).

Figure 2 illustrates the evolutionary dynamics for systems in which social learning evolves to an evolutionarily stable state (figure 2*a*), a locally stable state (figure 2*b*) or an evolutionary branching point (figure 2*c*).

## (a) Model extensions

In our basic model, the optimal social learning phenotype for a given forager depends on the state of the patch that the forager occupies. Foragers in low-occupancy patches benefit more from individual learning than those in high-occupancy patches (electronic supplementary material, figure S2). This can produce diversifying selection on the social learning phenotype. If a state-dependent (i.e. conditional) social learning rule can resolve this diversifying selection, then that rule may prevent the divergence of social learning phenotypes.

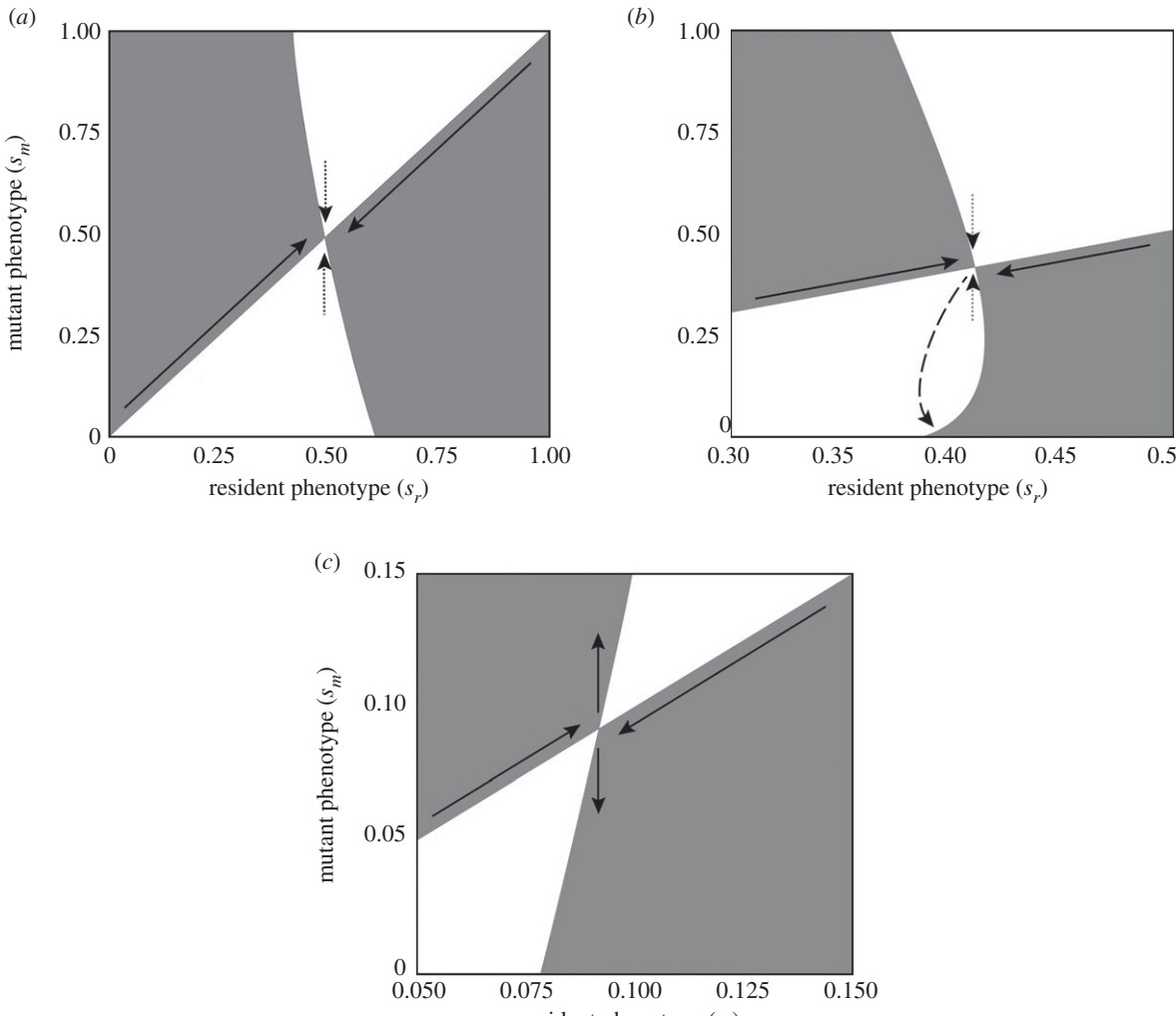

**Figure 2.** Pairwise invasibility plots for systems in which social learning evolves to globally (*a*) or locally (*b*) stable states or to an evolutionary branching point (*c*). Panels correspond to points A, B and C in figure 1. Grey (white) indicates that the mutant phenotype on the vertical axis can (cannot) invade the resident phenotype on the horizontal axis. Diagonal arrows show the direction of evolution in a monomorphic population, and vertical arrows show the direction of selection at the critical point where the monomorphic population stops evolving. The dashed arrow in (*b*) indicates that a sufficiently different mutant (in this case one with no social learning) can invade the resident population at the critical point. In all three panels, a population that begins with no social learning will evolve progressively greater social learning (along the upwards diagonal arrows) until it reaches a critical point at which a mutant with slightly greater social learning can no longer invade and replace the resident. In (*a*), no other social learning phenotype can invade the resident at this point. Thus, the critical point is evolutionarily stable. In (*b*), a slightly different mutant cannot invade the resident population, but a sufficiently different mutant can invade. Thus, the critical point is locally but not globally stable, and social learning phenotypes in the population may diverge. In (*c*), mutants with slightly higher or slightly lower social learning can invade the resident population. Thus, the critical point is an evolutionary branching point, and we expect the social learning phenotype to diverge in the population. In (*b*) and (*c*), we show only part of the resident axis in order to focus on the area near the critical point.

To understand whether this is likely to happen, we studied three simple but plausible conditional social learning rules.

(1) *Resource dependent*. Foragers use social learning if their current rate of resource collection is below some threshold, and use individual learning if their rate of resource collection is above that threshold. If a forager collects resources at exactly its threshold rate, then it uses social learning with a constant probability. The meaningful thresholds are the resource collection rates that foragers can achieve in the model. Ranked from highest to lowest, these are $1^{-q}, 2^{-q}, \ldots, n_{\max}^{-q}$ and 0. If $h_t$ is the position of a forager's social learning threshold in the ranked list of possible thresholds and $h_s$ is the probability that the forager uses social learning if it collects resources at exactly its threshold rate, then $h = h_t - h_s$ is continuous from 0 to $n_{\max} + 1$, and the conditions under

which foragers use social learning become more restricted as $h$ increases. We treat $h$ as an evolving trait.

(2) *Occupancy dependent*. Foragers use social learning if their patch is occupied by more than a threshold number of foragers, and use individual learning if their patch is occupied by fewer than that number of foragers. If their patch is occupied by exactly the threshold number of foragers, then they use social learning with a constant probability. Rule 2 differs from rule 1 only in how foragers learn when they occupy bad patches (i.e. patches with no resources). If $f_t$ is the number of occupants above which a forager always uses social learning and $f_s$ is the probability the forager uses social learning if there are exactly $f_t$ occupants in its patch, then $f = f_t - f_s$ is continuous from 0 to $n_{\max}$, and the conditions under which foragers use social learning become more restricted as $f$ increases. We treat $f$ as an evolving trait.

*Proc. R. Soc. B* **287**: 20192770

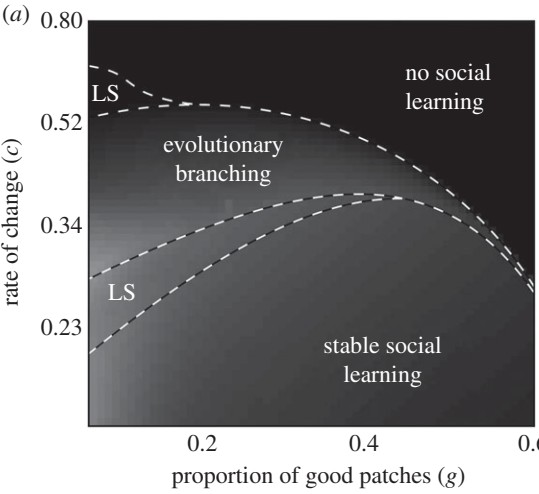
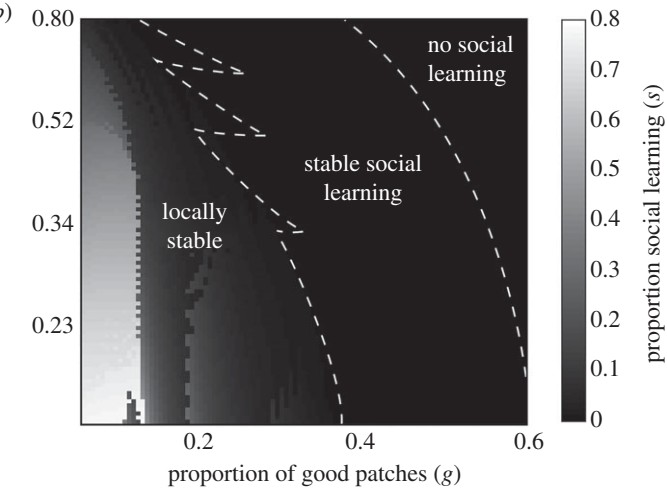

**Figure 3.** The proportion and evolutionary stability of social learning that evolves at different combinations of environmental heterogeneity and environmental change under conditional social learning rules 1 (*a*) and 2 (*b*). In (*a*), LS indicates the part of parameter space where the evolved social learning phenotype is locally stable.

(3) *Resource and occupancy dependent.* Foragers have separate occupancy-dependent rules for good patches (governed by $f_g$) and bad patches (governed by $f_b$). We allow $f_g$ and $f_b$ to evolve independently.

An important class of conditional social learning rules arises when each forager's social learning probability is itself a learned trait. These have been extensively studied elsewhere (e.g. [18,26,37,38]), and we will not study them further here.

As with the basic model, we studied the evolution of social learning rules 1–3 using the adaptive dynamics framework (see the electronic supplementary material, S4 for algorithms). We modelled systems with combinations of $c \in [0.15, 0.80]$ and $g \in [0.05, 0.60]$. For each system, we report the proportion of all learning that is social learning when the social learning rule is at its convergence-stable state and the population is at its steady state distribution. Furthermore, we report whether each convergence-stable state is globally stable, locally stable or an evolutionary branching point.

## (b) Results of model extensions

Under rule 1, social learning phenotypes can diverge, and the conditions that promote divergence are similar to those in the basic model (figure 3*a*). Social learning phenotypes diverge when expressing mutant phenotypes tends to concentrate foragers in patches with greater resource collection rates (electronic supplementary material, S5). In particular, if patch qualities change quickly, then unoccupied bad patches frequently become good. Mutants that express more individual learning are at an advantage because they can find these patches quickly. At the same time, there are many foragers in patches that have recently become bad, and mutants with more social learning are at an advantage because they more quickly escape bad patches. Thus, social learning phenotypes diverge (electronic supplementary material, figure S4). By contrast, if patch qualities change slowly, then most foragers become concentrated in densely occupied good patches. There are few empty good patches to find, so mutants with more individual learning gain no advantage. Foragers that use more social learning tend to discover patches that are little better than the crowded patches they already occupy, so mutants with more social learning gain no advantage. Thus, the convergence-stable social

phenotype is evolutionarily stable (electronic supplementary material, figure S5).

Under rule 2, the convergence-stable social learning phenotype is always at least locally stable, but social learning phenotypes may diverge if large mutations are possible (figure 3*b*). Under this rule, the optimal occupancy threshold at which social learning should start is different in good and bad patches. Foragers in bad patches can benefit by finding good patches even if they are crowded, but foragers in good patches benefit only if they find less crowded good patches. So, foragers in bad patches should use social learning at lower occupancies than foragers in good patches. The occupancy threshold for social learning settles between the optima for the two patch types. On average, good patches have higher occupancies than bad patches. This is true because foragers begin leaving a patch as soon as it turns bad, so any given patch will be less occupied when it is bad than when it was good (electronic supplementary material, table S1). Because of this, if a new phenotype with a higher occupancy threshold for social learning arises, the foragers that switch to individual learning will be disproportionately in good patches, because good patches are more crowded. Similarly, if a new phenotype with a lower occupancy for social learning arises, the foragers that switch to social learning will be disproportionately in bad patches, because bad patches are less crowded. Because divergence increases individual learning in good patches and increases social learning in bad patches, it is adaptive.

Under rule 3, different thresholds for social learning evolve in good and bad patches (figure 4). In every case we studied, the convergence-stable pair of thresholds was globally stable against invasions. This is because the population can evolve the optimal occupancy threshold for each patch type.

## 5. Discussion

Our results show that foraging competition in spatially and temporally variable environments can favour the divergence of social learning phenotypes, so that some foragers use social learning more often than others. This is true when foragers' social learning rates are independent of their current environments and also for simple conditional social learning rules.

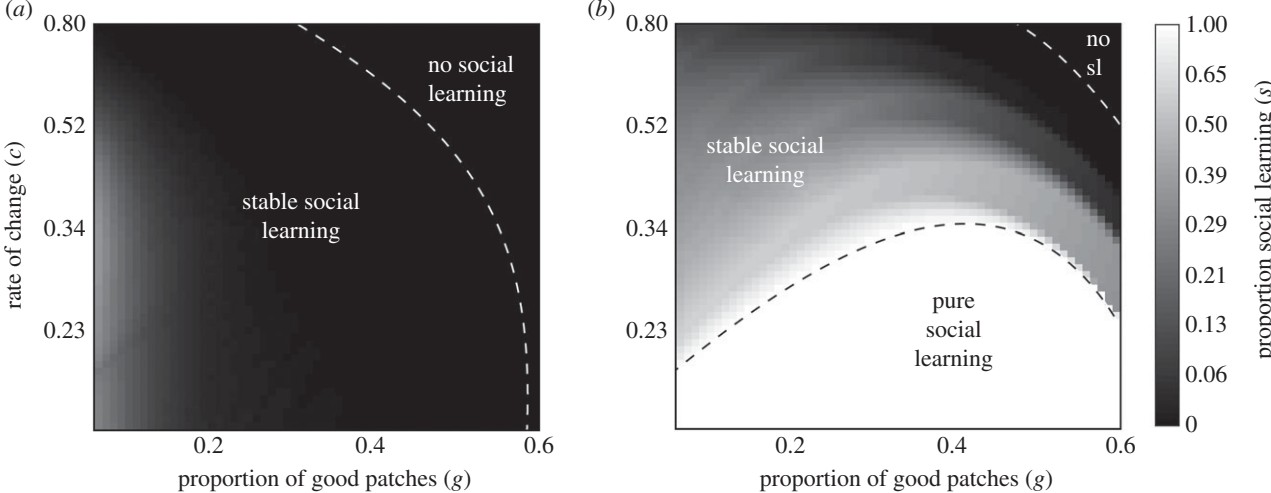

**Figure 4.** The proportion of social learning in good (a) and bad (b) patches that evolves at different combinations of environmental heterogeneity and environmental change under conditional social learning rule 3. Evolved social learning phenotypes were globally stable. Thus, we expect no divergence of the social learning phenotype.

This provides a mechanistic explanation for the divergence of social learning phenotypes that has been observed in nature (e.g. [15,20]). Interestingly, sufficiently complex conditional social learning rules can resolve disruptive selection and so may prevent the divergence of social learning phenotypes.

Our model is similar to classic producer-scrounger models [24]. Individual learning can find undiscovered resources and is similar to producing, while social learning finds only resources that have already been discovered by others and is similar to scrounging. Classic producer-scrounger models have not predicted the divergence of social learning phenotypes [23]. The critical difference between our model and classic producer-scrounger models is that, in classic producer-scrounger models, resources are consumed instantly when they are discovered. Thus, all foragers are always searching for new resources, and at any point in time every forager experiences exactly the same conditions. In our model, foragers exploit resources after they discover them. The optimal behaviour for each forager depends on the resources it has recently discovered, and these are different for different members of the population. In nature, foragers exploit resources after they find them. Thus, our model captures an important attribute of real systems that has been absent from much past theory. Interestingly, in a model with producers and scroungers where the time required for resource consumption was explicitly modelled, Hamblin & Giraldeau [37] observed cases in which social learning phenotypes did diverge. As in our model, this occurred when the overall frequency of social learning in the population was low. However, the divergence of social learning phenotypes was not the focus of Hamblin & Giraldeau's study, and the authors did not attempt to explain it or to understand the conditions that promote or prevent it. Thus, we do not know if this was a general attribute of their model, or whether it occurred only for the specific combination of parameter values they studied.

We have identified the part of parameter space in which social learning phenotypes are expected to diverge, but we have made no attempt to predict whether particular natural systems occupy that space. However, the part of parameter space that leads to the divergence of social learning phenotypes falls between the parts that lead to pure individual learning and to evolutionarily stable social learning. Thus, if environments exist that promote individual learning, and if

others exist that promote social learning, then environments that promote the divergence of social learning phenotypes should also exist. Recent work by Aplin & Morand-Ferron [20] found divergent use of social learning in each of five great tit (*Parus major*) populations. This suggests that conditions which lead to the divergence of social learning phenotypes may not be uncommon for some species.

The patches in our model might represent foraging sites, or they might represent different techniques that foragers can use (e.g. nut-hammering or termite-fishing in chimpanzees, [32]). Our model assumes that the number of sites or techniques available to foragers is large. This may be appropriate for foragers with large ranges or flexible foraging behaviours. However, in some systems, there may be only a small number of patches or techniques available to foragers. Our model cannot be easily modified to simulate systems with few foraging options, and new approaches are needed to test whether our results hold in such systems.

Social learning has been most studied in animals that are believed to be cognitively advanced [2,39]. Interestingly, in our model, sufficiently complex conditional social learning rules can resolve disruptive selection and prevent the divergence of social learning phenotypes. Complex social learning strategies may be most achievable by cognitively advanced animals. If this is true, then the divergence of social learning phenotypes may not be a hallmark of cognitive advancement, but rather an alternative to it. We believe that studies of social learning phenotypes across a broader taxonomic range, including species with limited cognition, would do much to advance our understanding of how social learning evolves and the conditions under which animal cultures emerge.

Data accessibility. MATLAB codes for models presented in this paper are available from the Dryad Digital repository: https://doi.org/10.5061/dryad.v6wwpzgrm [36].

Competing interests. We declare we have no competing interests.

Authors' contributions. R.T.G. and M.S. conceived the study; R.T.G. and F.J. constructed the model and analysed the results; all authors wrote the manuscript.

Funding. R.T.G. was partially supported by Natural Environment Research Council grant no. NE/L002469/1. M.S. was supported by a grant to Erol Akçay from the Army Research Office (grant no. W911NF-17-1-0017).

Acknowledgements. The authors thank B.B. Chapman for comments.

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
