## [Reviewer comments · Proceedings of the Royal Society B: Biological Sciences]

Review History

RSPB-2019-1276.R0 (Original submission)

Review form: Reviewer 1

Recommendation

Major revision is needed (please make suggestions in comments)

Scientific importance: Is the manuscript an original and important contribution to its field?

Acceptable

General interest: Is the paper of sufficient general interest?

Marginal

Quality of the paper: Is the overall quality of the paper suitable?

Acceptable

Is the length of the paper justified?

Yes

Should the paper be seen by a specialist statistical reviewer?

No

Do you have any concerns about statistical analyses in this paper? If so, please specify them explicitly in your report.

No

It is a condition of publication that authors make their supporting data, code and materials available - either as supplementary material or hosted in an external repository. Please rate, if applicable, the supporting data on the following criteria.

Is it accessible?

N/A

Is it clear?

N/A

Is it adequate?

N/A

Do you have any ethical concerns with this paper?

No

Comments to the Author

****This review uses Markdown formatting; using a software for rendering markdown documents will facilitate reading through this review.****

Review of "Competition for resources can promote the divergence of social learning phenotypes"

Description of the work

The manuscript describes a theoretical model of the evolution of social and individual learning in a foraging task. One of the authors' goal is to determine whether they can reproduce the results of a previous model regarding the divergence of social learning phenotypes. They use a deterministic setting to ensure that divergence (or polymorphism) is not due to stochastic effects.

The model assumes that there is a large number of patches and allows individuals to move between these patches so as to maximize resource consumption, under the constraint of a genetically determined decision rule (probability of social learning used). The authors let evolve the probability of social learning and identify regions of the parameter space where natural selection leads to different outcomes (evolutionary stability of social learning, branching, local stability, absence of social learning). The authors also examine to what extent the possibility to display conditional social learning strategies impedes the evolutionary branching of social learning phenotypes.

Major comments and recommendation

Contribution

The manuscript is an interesting contribution in that it demonstrates well how a rather simple foraging task can lead to polymorphism in social learning types. Also, it touches upon an interesting and complicated question which is to predict whether organismal complexity is going to evolve rather than divergence of simpler organisms. In particular, it shows that adding one layer of organismal complexity is not always sufficient to impede divergence and polymorphism.

Issues

Complexity of the model

Even though the verbal description of the model is clear and follows from the introduction, I was wondering whether the model could not be simplified and still allow the authors to address their question of interest. For instance, would a model with only two patches (one good and one bad) still capture the main effects outlined in the manuscript? I mention this especially in view of the fact that no analytical results were provided by the authors, which is indeed a consequence of the complexity of the model's assumptions. It would be of great added value to have analytical results in a baseline model.

Clarity of the presentation of the mathematical model

- Boxes 1 and 2 poorly explain how the equations are derived and contain notational issues that make them quite difficult to follow (by the way, I would rather put the content of boxes 1 and 2 in an Appendix or a Supplementary information). The way the authors "update" the matrix M (by calling it with a different symbol, A) is probably how they programmed the model in their code but can't be used to describe it in the manuscript. Indeed when the authors write in equation B1.4 that $a_{[k-1,k]} = a_{[k-1,k]} + X$, this is simply wrong (simplifying leads to $X=0$...). I appreciate the fact that the authors wanted to decompose the different terms of the matrix entries, but there are many other ways to do so. For instance, they could write $M_{[k-1,k]} = L_S + L_I$ where L_S is the loss of foragers after social learning and L_I is the loss of foragers after individual learning. Of course, they should stop using A and B instead of M and N , this just brings additional confusion.

- Often, the order in which the factors/terms are described in the text of the boxes does not correspond to the order in which the factors/terms occur in the equation that is being explained (for instance take equation B1.5 and its explanation above it).

- In the model extensions, are the threshold t and f evolving traits or parameters? It was not clear from the model description, even though they look to me as parameters. But then the figures (which are not numbered...) describing the model extensions do not state the values of f and t . If these are parameters, is s_t (s_f resp.) the evolving trait? The authors say that they track the proportion of social learning in the population $s \cdot$ (even this notation is not very transparent), which is an average taken over the different conditions an individual with a given threshold might find itself in. This is probably not the evolving trait itself, if I understand correctly. Please clarify all of the above.

Explanation of model results

Even though I appreciate the authors' attempt at providing intuitive explanations to their results, these explanations are not sufficient if unsupported by quantitative facts. For instance, on lines 237--247, the authors make a number of claims for which we would like to have proof. I realize that the current numerical analysis of their abstract model would not allow them supporting these claims. Consequently, the authors may consider performing individual-based simulations in order to obtain temporal data regarding the behavior of individual foragers.

Emphasis on animal personality

The authors model only one type of task or context (i.e., foraging) but make statements about the relevance of their model to understand the evolution of animal personality. I think the link is tenuous, and I don't see the added value to discussing animal personality here. The authors only show conditions under which social learning can diverge. The further evolution of animal personality requires capturing other phenomena not described by the authors.

Recommendation

Although the paper is potential interest, I think the authors need to address the issues mentioned in my review before this paper can be accepted for publication.

Minor comments

- The choice of terminology could be discussed: why talking about divergence rather than polymorphisms, especially given that the authors oppose the concept of divergence to monomorphism?
- At several instances the authors make the mistake of calling a phenotype monomorphic. The standard way these words are used in evolutionary biology is to say that a population is either monomorphic (containing individuals of only one given phenotype or "morph") or polymorphic (many co-existing phenotypes in the population).
- l. 130: I find the word "assays" not appropriate for the context. I would rephrase using the concept of trait-substitution sequence.
- l. 198: "or 10"  "of 10"
- l. 223: "by replicating the results"  "by replicating the parameter combinations" or "by replicating the environmental conditions".
- l. 272--274: weird modeling choice: why not use a general function for s_t that increases as a function of t ?
- l. 423--424: How can you draw this conclusion only from your model and one empirical study?
- l. 439--442: " Such complex strategies may be most achievable by cognitively advanced animals. If this is true, then the divergence of social learning phenotypes may not be a hallmark of cognitive advancement, but rather an alternative to it.": This statement depends a lot on the limitations/constraints of the cognitive system and also depends on the the complexity of the task to be performed.
- l. 529: Remove "additive", a term is additive by definition (for elements of multiplications, we use "factor").

I hope these comments will be useful to the authors.

With my best regards.

Review form: Reviewer 2

Recommendation

Accept with minor revision (please list in comments)

Scientific importance: Is the manuscript an original and important contribution to its field?

Good

General interest: Is the paper of sufficient general interest?

Acceptable

Quality of the paper: Is the overall quality of the paper suitable?

Good

Is the length of the paper justified?

Yes

Should the paper be seen by a specialist statistical reviewer?

Yes

Do you have any concerns about statistical analyses in this paper? If so, please specify them explicitly in your report.

No

It is a condition of publication that authors make their supporting data, code and materials available - either as supplementary material or hosted in an external repository. Please rate, if applicable, the supporting data on the following criteria.

Is it accessible?

N/A

Is it clear?

N/A

Is it adequate?

N/A

Do you have any ethical concerns with this paper?

No

Comments to the Author

It is an interesting manuscript and I enjoyed reading it, despite I don't really understand the equations (and I recommend the editor to seek a reviewer with enough math background to check those). The authors tested which rates of social learning were beneficial in different circumstances and found that when both the speed of environmental change and environmental heterogeneity were high, the probability of social learning evolved to an evolutionary branching point, where selection favoured the diversification of social learning probabilities (i.e. both higher and lower rates of social learning were beneficial). This is an important finding, and the authors did a good job of explaining and discussing this result in a way that makes sense to a biologist. Therefore, I believe that the manuscript has a potential to contribute to the current studies of social learning and cultural evolution.

There are however some weaknesses that I have to mention. First of all, what confuses me is that the authors did not actually observe the divergence of the learning phenotypes in their model, but only the conditions that made this divergence likely to occur. It is probably somehow related to the way they built and implemented their model, but there are not enough details (at least for me) in the manuscript to assess that. Maybe it would be helpful to describe how the model has been implemented (using what software and building blocks), and explain why it was not possible to observe the actual divergence of phenotypes.

Also, I strongly feel that the reference to the development of personality does not belong to this manuscript. Social learning has very little to do with animal personality, which is the combination of many different traits, many of them being innate. The reference to the development of personality looks alien in this (otherwise pretty good) manuscript and makes me think that it was included mainly for the purpose of making the manuscript more attractive. If you need something catchy, I would rather make an emphasis on the implications to the cultural evolution instead.

In terms of interpretation and discussion of the results, it would be helpful to give a simple example of real-life divergence of social learning phenotypes driven by the environment alone,

without pre-existing differences among individuals. The authors refer to Aplin and Morand-Ferron (2017) but do not give any examples from there, which makes it difficult to visualize it to someone who (like myself) do not have time or desire to read that additional paper. And a small final remark: male sperm whales do not fail to learn coda vocalizations (as you claim on line 38). They just do not produce them, as these vocalizations are mostly used for intra-group communication, and adult males do not belong to any group.

Decision letter (RSPB-2019-1276.R0)

12-Jul-2019

Dear Dr Gilman:

I am writing to inform you that your manuscript RSPB-2019-1276 entitled "Competition for resources can promote the divergence of social learning phenotypes" has, in its current form, been rejected for publication in Proceedings B.

This action has been taken on the advice of referees, who have recommended that substantial revisions are necessary. In particular, Reviewer 1 expresses some concern about the model, and as Reviewer 2 points out, this paper will be challenging to follow for readers not familiar with the math. I think the suggestion to include more relevant published examples is excellent. I also agree with the reviewers that the extension to personality is a stretch. I see no problem with mentioning it, but I think it needs to be restricted to a possible future direction of the work. If these changes can be addressed, we would be happy to consider a resubmission. However please note that this is not a provisional acceptance.

Sincerely,
Dr Sarah Brosnan
Editor, Proceedings B
mailto: proceedingsb@royalsociety.org

Associate Editor

Comments to Author:

I thank the authors for their submission. I have now received the feedback from two expert reviewers. Both agree, and so do I, that the study has the potential to appeal to broad readership of Proceedings B, while also providing a novel contribution to the field. However, both reviewers have raised significant concerns which would need to be addressed before the manuscript could be considered further. I would recommend that any revision be sent out again for review by a modelling expert given the significant mathematical concerns (and potentially mistakes) raised by Reviewer 1.

Reviewer(s)' Comments to Author:

Referee: 1

Comments to the Author(s)

This review uses Markdown formatting; using a software for rendering markdown documents will facilitate reading through this review.

Review of "Competition for resources can promote the divergence of social learning phenotypes"

Description of the work

The manuscript describes a theoretical model of the evolution of social and individual learning in a foraging task. One of the authors' goal is to determine whether they can reproduce the results of a previous model regarding the divergence of social learning phenotypes. They use a deterministic setting to ensure that divergence (or polymorphism) is not due to stochastic effects.

The model assumes that there is a large number of patches and allows individuals to move between these patches so as to maximize resource consumption, under the constraint of a genetically determined decision rule (probability of social learning used). The authors let evolve the probability of social learning and identify regions of the parameter space where natural selection leads to different outcomes (evolutionary stability of social learning, branching, local stability, absence of social learning). The authors also examine to what extent the possibility to display conditional social learning strategies impedes the evolutionary branching of social learning phenotypes.

Major comments and recommendation

Contribution

The manuscript is an interesting contribution in that it demonstrates well how a rather simple foraging task can lead to polymorphism in social learning types. Also, it touches upon an interesting and complicated question which is to predict whether organismal complexity is going to evolve rather than divergence of simpler organisms. In particular, it shows that adding one layer of organismal complexity is not always sufficient to impede divergence and polymorphism.

Issues

Complexity of the model

Even though the verbal description of the model is clear and follows from the introduction, I was wondering whether the model could not be simplified and still allow the authors to address their question of interest. For instance, would a model with only two patches (one good and one bad) still capture the main effects outlined in the manuscript? I mention this especially in view of the fact that no analytical results were provided by the authors, which is indeed a consequence of the complexity of the model's assumptions. It would be of great added value to have analytical results in a baseline model.

Clarity of the presentation of the mathematical model

- Boxes 1 and 2 poorly explain how the equations are derived and contain notational issues that make them quite difficult to follow (by the way, I would rather put the content of boxes 1 and 2 in an Appendix or a Supplementary information). The way the authors "update" the matrix M (by calling it with a different symbol, A) is probably how they programmed the model in their code but can't be used to describe it in the manuscript. Indeed when the authors write in equation B1.4 that $a_{[k-1,k]} = a_{[k-1,k]} + X$, this is simply wrong (simplifying leads to $X=0$...). I appreciate the fact that the authors wanted to decompose the different terms of the matrix entries, but there are many other ways to do so. For instance, they could write $M_{[k-1,k]} = L_S + L_I$ where L_S is the loss of foragers after social learning and L_I is the loss of foragers after individual learning. Of course, they should stop using A and B instead of M and N , this just brings additional confusion.

- Often, the order in which the factors/terms are described in the text of the boxes does not correspond to the order in which the factors/terms occur in the equation that is being explained (for instance take equation B1.5 and its explanation above it).

- In the model extensions, are the threshold t and f evolving traits or parameters? It was not clear from the model description, even though they look to me as parameters. But then the figures (which are not numbered...) describing the model extensions do not state the values of f and t . If these are parameters, is s_t (s_f resp.) the evolving trait? The authors say that they track the proportion of social learning in the population $s \cdot$ (even this notation is not very transparent), which is an average taken over the different conditions an individual with a given threshold might find itself in. This is probably not the evolving trait itself, if I understand correctly. Please clarify all of the above.

Explanation of model results

Even though I appreciate the authors' attempt at providing intuitive explanations to their results, these explanations are not sufficient if unsupported by quantitative facts. For instance, on lines 237--247, the authors make a number of claims for which we would like to have proof. I realize that the current numerical analysis of their abstract model would not allow them supporting these claims. Consequently, the authors may consider performing individual-based simulations in order to obtain temporal data regarding the behavior of individual foragers.

Emphasis on animal personality

The authors model only one type of task or context (i.e., foraging) but make statements about the relevance of their model to understand the evolution of animal personality. I think the link is tenuous, and I don't see the added value to discussing animal personality here. The authors only show conditions under which social learning can diverge. The further evolution of animal personality requires capturing other phenomena not described by the authors.

Recommendation

Although the paper is potential interest, I think the authors need to address the issues mentioned in my review before this paper can be accepted for publication.

Minor comments

- The choice of terminology could be discussed: why talking about divergence rather than polymorphisms, especially given that the authors oppose the concept of divergence to monomorphism?

- At several instances the authors make the mistake of calling a phenotype monomorphic. The standard way these words are used in evolutionary biology is to say that a population is either monomorphic (containing individuals of only one given phenotype or "morph") or polymorphic (many co-existing phenotypes in the population).

- l. 130: I find the word "assays" not appropriate for the context. I would rephrase using the concept of trait-substitution sequence.

- l. 198: "or 10"  "of 10"

- l. 223: "by replicating the results"  "by replicating the parameter combinations" or "by replicating the environmental conditions".

- l. 272--274: weird modeling choice: why not use a general function for s_t that increases as a function of t ?

- l. 423--424: How can you draw this conclusion only from your model and one empirical study?

- l. 439--442: "Such complex strategies may be most achievable by cognitively advanced animals. If this is true, then the divergence of social learning phenotypes may not be a hallmark of cognitive advancement, but rather an alternative to it.": This statement depends a lot on the limitations/constraints of the cognitive system and also depends on the the complexity of the task to be performed.

-l. 529: Remove "additive", a term is additive by definition (for elements of multiplications, we use "factor").

I hope these comments will be useful to the authors.

With my best regards.

Referee: 2

Comments to the Author(s)

It is an interesting manuscript and I enjoyed reading it, despite I don't really understand the equations (and I recommend the editor to seek a reviewer with enough math background to check those). The authors tested which rates of social learning were beneficial in different circumstances and found that when both the speed of environmental change and environmental heterogeneity were high, the probability of social learning evolved to an evolutionary branching point, where selection favoured the diversification of social learning probabilities (i.e. both higher and lower rates of social learning were beneficial). This is an important finding, and the authors did a good job of explaining and discussing this result in a way that makes sense to a biologist.

Therefore, I believe that the manuscript has a potential to contribute to the current studies of social learning and cultural evolution.

There are however some weaknesses that I have to mention. First of all, what confuses me is that the authors did not actually observe the divergence of the learning phenotypes in their model, but only the conditions that made this divergence likely to occur. It is probably somehow related to the way they built and implemented their model, but there are not enough details (at least for me) in the manuscript to assess that. Maybe it would be helpful to describe how the model has been implemented (using what software and building blocks), and explain why it was not possible to observe the actual divergence of phenotypes.

Also, I strongly feel that the reference to the development of personality does not belong to this manuscript. Social learning has very little to do with animal personality, which is the combination of many different traits, many of them being innate. The reference to the development of personality looks alien in this (otherwise pretty good) manuscript and makes me think that it was included mainly for the purpose of making the manuscript more attractive. If you need something catchy, I would rather make an emphasis on the implications to the cultural evolution instead.

In terms of interpretation and discussion of the results, it would be helpful to give a simple example of real-life divergence of social learning phenotypes driven by the environment alone, without pre-existing differences among individuals. The authors refer to Aplin and Morand-Ferron (2017) but do not give any examples from there, which makes it difficult to visualize it to someone who (like myself) do not have time or desire to read that additional paper.

And a small final remark: male sperm whales do not fail to learn coda vocalizations (as you claim on line 38). They just do not produce them, as these vocalizations are mostly used for intra-group communication, and adult males do not belong to any group.

Author's Response to Decision Letter for (RSPB-2019-1276.R0)

See Appendix A.

RSPB-2019-2770.R0

Review form: Reviewer 1

Recommendation

Accept with minor revision (please list in comments)

Scientific importance: Is the manuscript an original and important contribution to its field?

Good

General interest: Is the paper of sufficient general interest?

Acceptable

Quality of the paper: Is the overall quality of the paper suitable?

Good

Is the length of the paper justified?

Yes

Should the paper be seen by a specialist statistical reviewer?

No

Do you have any concerns about statistical analyses in this paper? If so, please specify them explicitly in your report.

No

It is a condition of publication that authors make their supporting data, code and materials available - either as supplementary material or hosted in an external repository. Please rate, if applicable, the supporting data on the following criteria.

Is it accessible?

No

Is it clear?

No

Is it adequate?

N/A

Do you have any ethical concerns with this paper?

No

Comments to the Author

Congratulations, you made significant improvements to the paper. Please provide all code used to produce the results in the paper, so that readers can reproduce your findings as easily as possible.

Decision letter (RSPB-2019-2770.R0)

27-Jan-2020

Dear Dr Gilman

I am pleased to inform you that your manuscript RSPB-2019-2770 entitled "Competition for resources can promote the divergence of social learning phenotypes" has been accepted for publication in Proceedings B.

The referee(s) have recommended publication, with the condition that you provide the complete code for your model in a clear and usable format. Because the schedule for publication is very tight, it is a condition of publication that you submit the revised version of your manuscript within 7 days. If you do not think you will be able to meet this date please let us know.

When submitting your revised manuscript, you will be able to respond to the comments made by

the referee(s) and upload a file "Response to Referees". You can use this to document any changes you make to the original manuscript. We require a copy of the manuscript with revisions made since the previous version marked as 'tracked changes' to be included in the 'response to referees' document.

Sincerely,
Dr Sarah Brosnan
Editor, Proceedings B
mailto: proceedingsb@royalsociety.org

Reviewer(s)' Comments to Author:

Referee: 1

Comments to the Author(s).

Congratulations, you made significant improvements to the paper. Please provide all code used to produce the results in the paper, so that readers can reproduce your findings as easily as possible.

Decision letter (RSPB-2019-2770.R1)

29-Jan-2020

Dear Dr Gilman

I am pleased to inform you that your manuscript entitled "Competition for resources can promote the divergence of social learning phenotypes" has been accepted for publication in Proceedings B.

Your article has been estimated as being 9 pages long. Our Production Office will be able to confirm the exact length at proof stage.

Open Access

Paper charges

Sincerely,
Proceedings B
mailto:proceedingsb@royalsociety.org

Appendix A

C.1249a Michael Smith Building
Dept. of Earth and Environmental Sciences
University of Manchester
Manchester, UK
26 November 2019

Dr. Sarah Brosnan
Editor
Proceedings of the Royal Society B – Biological Sciences

Dear Prof. Brosnan,

We are very grateful to you, the associate editor and the anonymous reviewers for your helpful comments on our manuscript (RSPB-2019-1276; Competition for resources can promote the divergence of social learning phenotypes). We have revised the manuscript in response to the comments, and we believe it is much improved as a result.

With this letter, we resubmit the manuscript. Our responses to the reviewer comments are embedded in the comments below (in green so they can be easily identified). The line numbers in the responses are those in the revised manuscript with track changes showing.

We thank you again for your consideration.

Sincerely,

R. Tucker Gilman, Fern Johnson and Marco Smolla

Reviewer(s)' Comments to Author:

Referee: 1

Comments to the Author(s)

This review uses Markdown formatting; using a software for rendering markdown documents will facilitate reading through this review.

Review of "Competition for resources can promote the divergence of social learning phenotypes"

Description of the work

The manuscript describes a theoretical model of the evolution of social and individual learning in a foraging task. One of the authors' goal is to determine whether they can reproduce the results of a previous model regarding the divergence of social learning phenotypes. They use a deterministic setting to ensure that divergence (or polymorphism) is not due to stochastic effects.

The model assumes that there is a large number of patches and allows individuals to move between these patches so as to maximize resource consumption, under the constraint of a genetically determined decision rule (probability of social learning used). The authors let evolve the probability of social learning and identify regions of the parameter space where natural selection leads to different outcomes (evolutionary stability of social learning, branching, local stability, absence of social learning). The authors also examine to what extent the possibility to display conditional social learning strategies impedes the evolutionary branching of social learning phenotypes.

Major comments and recommendation

Contribution

The manuscript is an interesting contribution in that it demonstrates well how a rather simple foraging task can lead to polymorphism in social learning types. Also, it touches upon an interesting and complicated question which is to predict whether organismal complexity is going to evolve rather than divergence of simpler organisms. In particular, it shows that adding one layer of organismal complexity is not always sufficient to impede divergence and polymorphism.

Issues

Complexity of the model

Even though the verbal description of the model is clear and follows from the introduction, I was wondering whether the model could not be simplified and still allow the authors to address their question of interest. For instance, would a model with only two patches (one good and one bad) still capture the main effects outlined in the manuscript? I mention this especially in view of the fact that no analytical results were provided by the authors, which is indeed a consequence of the complexity of the model's assumptions. It would be of great added value to have analytical results in a baseline model.

We thank the reviewer for this comment. Like the reviewer, our early intuition was that a two-patch version of our model would be useful. However, when we sat down to build it, we found that the two-patch model is surprisingly difficult to analyse. The reason is that, in the two-patch model, there is no steady-state distribution for foragers among patches. For example, if the system has two good patches and one of those patches turns bad, foragers will move from the bad patch to the good patch over time, so the distribution of foragers between the patches is not constant. If we know the distribution of foragers when the good patch turns bad (e.g., if foragers are at the ideal free distribution), then we can write down the rate at which foragers move from the bad to the good patch as a function of time from the change, and we can solve the ODE for the expected density of foragers in each patch over time. Then, we can write down the relative fitness of different learning types after the change as an integral over time. But, we cannot solve this integral in closed form. Moreover, even if we could solve the integral in closed form, we would face another, bigger problem. In general, we *do not* know the distribution of foragers in the good patches when one patch turns bad. This distribution depends on how long the patches were both good. If they were good for only a short time, then the population will not have achieved the ideal free distribution by the time of the change, and one patch will have more foragers than the other. So, to determine the relative fitness of different learning strategies after a change in patch quality, we would have to integrate across all of the different ways the patches might have reached the state they transitioned from, and across all of the different times that they might have occupied that state. Moreover, we would have to do the same thing for the state before that, and the state before that, *ad infinitum*. In contrast, our assumption that the system of patches is large allows us

to find a steady-state distribution of foragers (albeit numerically), and to study the fitness of new social learning phenotypes at the steady-state distribution.

Nonetheless, the reviewer's comment highlights an important point. We studied a system with many different foraging options, and we cannot be confident that our results will hold in systems with very small numbers of foraging options. We now discuss this in lines 413-421, and mention the limitation in lines 5-7 and 69.

Clarity of the presentation of the mathematical model

- Boxes 1 and 2 poorly explain how the equations are derived and contain notational issues that make them quite difficult to follow (by the way, I would rather put the content of boxes 1 and 2 in an Appendix or a Supplementary information). The way the authors "update" the matrix M (by calling it with a different symbol, A) is probably how they programmed the model in their code but can't be used to describe it in the manuscript. Indeed when the authors write in equation B1.4 that $a_{k-1,k} = a_{k-1,k} + X$, this is simply wrong (simplifying leads to $X=0$...). I appreciate the fact that the authors wanted to decompose the different terms of the matrix entries, but there are many other ways to do so. For instance, they could write $M_{k-1,k} = L_S + L_I$ where L_S is the loss of foragers after social learning and L_I is the loss of foragers after individual learning. Of course, they should stop using A and B instead of M and N , this just brings additional confusion.

The reviewer is absolutely correct. Happily, the errors the reviewer points out were in the presentation, not in the model itself. We have rewritten these sections, and moved them to the supplemental material (sections S1 and S2) as the reviewer suggested. The result is both shorter and (we think) clearer. We thank the reviewer for the advice!

- Often, the order in which the factors/terms are described in the text of the boxes does not correspond to the order in which the factors/terms occur in the equation that is being explained (for instance take equation B1.5 and its explanation above it).

We have corrected this in the rewritten sections.

- In the model extensions, are the threshold t and f evolving traits or parameters? It was not clear from the model description, even though they look to me as parameters. But then the figures (which are not numbered...) describing the model extensions do not state the values of f and t . If these are parameters, is s_t (s_f resp.) the evolving trait? The authors say that they track the proportion of social learning in the population s (even this notation is not very transparent), which is an average taken over the different conditions an individual with a given threshold might find itself in. This is probably not the evolving trait itself, if I understand correctly. Please clarify all of the above.

We agree that the model extensions were not clearly explained, and we have rewritten the section (lines 285-324). Moreover, in keeping with the reviewer's advice for the boxes, we have moved the details of the algorithm for finding and testing convergence-stable states in the extended model to the supplemental materials (section S4). While the algorithms will be important to readers who wish to expand or replicate our study, they may not be of interest to readers who are primarily interested in our biological results. We think moving the algorithm to the supplemental materials will help address reviewer 2's concern that the originally submitted manuscript was too technical for some non-specialist readers.

Explanation of model results

Even though I appreciate the authors' attempt at providing intuitive explanations to their results, these explanations are not sufficient if unsupported by quantitative facts. For instance, on lines 237--247, the authors make a number of claims for which we would like to have proof. I realize that the current numerical analysis of their abstract model would not allow them supporting these claims. Consequently, the authors may consider performing individual-based simulations in order to obtain temporal data regarding the behavior of individual foragers.

We thank the reviewer for this very helpful advice. In our opinion, individual-based simulations are not ideal for illustrating the mechanisms that promote or prevent the divergence of social learning phenotypes. The reason is that individual-based simulations are necessarily stochastic, and in the presence of stochasticity it becomes very difficult to separate the diversifying effect of mutation and the stabilising effect of drift from the effects of selection that we wish to study. Thus, in our revision, we provide evidence to support our claims by studying our deterministic model directly. Because the evidence is lengthy and somewhat technical, we have placed it in the supplementary materials (sections S3 and S5), and we refer readers to those sections and the supporting figures at appropriate points throughout the results.

Emphasis on animal personality

The authors model only one type of task or context (i.e., foraging) but make statements about the relevance of their model to understand the evolution of animal personality. I think the link is tenuous, and I don't see the added value to discussing animal personality here. The authors only show conditions under which social learning can diverge. The further evolution of animal personality requires capturing other phenomena not described by the authors.

Both reviewers and the editor agreed that the connections to animal personality in our manuscript were tenuous, and we have removed most discussion of animal personality from the revision. We still acknowledge in the introduction that the divergence of social learning phenotypes can promote the divergence of other behavioural traits, and that this has the potential to lead to the evolution of personality. This is the result of Arbilly and colleagues' (2010) frequently cited model that studies the evolutionary effects social learning polymorphism (as opposed the evolutionary origins of social learning polymorphism, as we study here), and we believe it is relevant background. However, in our revision, we have made no conjectures about animal personality based on our own results. We no longer mention animal personality in the abstract, the study overview, or the discussion of our results.

Recommendation

Although the paper is potential interest, I think the authors need to address the issues mentioned in my review before this paper can be accepted for publication.

Minor comments

- The choice of terminology could be discussed: why talking about divergence rather than polymorphisms, especially given that the authors oppose the concept of divergence to monomorphism?

The ideas of divergence and polymorphism are closely linked: divergence is the process that leads to the state of polymorphism. We framed the paper in terms of divergence because we are more interested in processes than states. Indeed, our adaptive dynamics approach predicts divergence, and infers polymorphism from the fact that divergence has occurred. However, because we misused the term "monomorphic" in some places in our earlier submission (as the reviewer points out – see the next comment), we think this was more confusing than it needed to be. That error has now been corrected. Moreover, we think that we jumped from the idea of polymorphism to divergence without a clear transition in the introduction to our earlier submission. We have rewritten lines 58-60 and reworded throughout the introduction to make the transition easier to follow.

- At several instances the authors make the mistake of calling a phenotype monomorphic. The standard way these words are used in evolutionary biology is to say that a population is either monomorphic (containing individuals of only one given phenotype or "morph") or polymorphic (many co-existing phenotypes in the population).

We have corrected this error throughout the manuscript, and thank the reviewer for catching it.

- l. 130: I find the word "assays" not appropriate for the context. I would rephrase using the concept of trait-substitution sequence.

As we understand it, the trait substitution sequence refers to the process by which a mutant phenotype invades, increases, and eventually replaces a resident. We meant something slightly different by "invasion assay." By invasion assay, we meant the test of whether a particular mutant can or cannot invade a resident population. So, an invasion assay might or might not lead to a trait substitution sequence. Most classic adaptive dynamics models have not needed invasion assays, because they analytically obtain the derivatives of fitness with respect to the phenotypes of interest. We cannot obtain those derivatives analytically, and therefore we assessed the fitness of incremental mutations numerically. However, we agree with the reviewer that the term "invasion assay" is not common in the literature on adaptive dynamics, and we have reworded to avoid it.

- l. 198: "or 10"  "of 10"

Corrected (line 203).

- l. 223: "by replicating the results"  "by replicating the parameter combinations" or "by replicating the environmental conditions".

We agree that "replicating the results" was not the correct wording, and have reworded accordingly (line 229).

- l. 272--274: weird modeling choice: why not use a general function for s_t that increases as a function of t ?

The social learning frequency is indeed a monotonic function of the evolving trait, as the reviewer suggests it should be. However, this was poorly explained in our original submission. We have clarified this in the rewritten section on the model extensions (lines 285-324 and supplementary materials section S4).

- l. 423--424: How can you draw this conclusion only from your model and one empirical study?

We agree that this conclusion was poorly worded. We do not know if the conditions that promote the divergence of social learning phenotypes in our models are common in nature. However, Aplin and Morand-Ferron (2017) studied five great tit populations, and they found diverged social learning phenotypes in all five. So, for at least some species, the conditions that allow different social learning phenotypes to be maintained in the same population may not be uncommon. We have reworded the sentence accordingly (lines 410-412).

- l. 439--442: "Such complex strategies may be most achievable by cognitively advanced animals. If this is true, then the divergence of social learning phenotypes may not be a hallmark of cognitive advancement, but rather an alternative to it.": This statement depends a lot on the limitations/constraints of the cognitive system and also depends on the complexity of the task to be performed.

We believe the reviewer is questioning the statement, "Such complex strategies may be most achievable by cognitively advanced animals." If so, we agree. However, we are not arguing that this is true. We are arguing that, *if* this is true, then diversity may be an alternative to complexity. Even so, we agree that this is a *post hoc* hypothesis of our study, not a conclusion, and we have edited the paragraph to make that clearer (lines 425-431).

- l. 529: Remove "additive", a term is additive by definition (for elements of multiplications, we use "factor").

We have removed the word, and thank the reviewer for the correction.

I hope these comments will be useful to the authors.

With my best regards.

Referee: 2

Comments to the Author(s)

It is an interesting manuscript and I enjoyed reading it, despite I don't really understand the equations (and I recommend the editor to seek a reviewer with enough math background to check those). The authors tested which rates of social learning were beneficial in different circumstances and found that when both the speed of environmental change and environmental heterogeneity were high, the probability of social learning evolved to an evolutionary branching point, where selection favoured the diversification of social learning probabilities (i.e. both higher and lower rates of social learning were beneficial). This is an important finding, and the authors did a good job of explaining and discussing this result in a way that makes sense to a biologist. Therefore, I believe that the manuscript has a potential to contribute to the current studies of social learning and cultural evolution.

Thank you! We appreciate the reviewer's support.

There are however some weaknesses that I have to mention. First of all, what confuses me is that the authors did not actually observe the divergence of the learning phenotypes in their model, but only the conditions that made this divergence likely to occur. It is probably somehow related to the way they built and implemented their model, but there are not enough details (at least for me) in the manuscript to assess that. Maybe it would be helpful to describe how the model has been implemented (using what software and building blocks), and explain why it was not possible to observe the actual divergence of phenotypes.

In this manuscript, we analyse our model using tools from adaptive dynamics. Adaptive dynamics “works” by asking whether a rare mutant phenotype can invade a monomorphic resident population. If mutations have small effects and the resident phenotype is sufficiently far from certain critical points, a mutant that can invade the resident population can also replace the resident. So, by studying a sequence of trait substitutions, we can infer the evolutionary trajectory of the system. When the system reaches a convergence-stable point, we may find that no mutation can invade the resident population, in which case the convergence-stable point is also evolutionarily stable. Alternatively, we may find that both mutations that increase *and* mutations that decrease the phenotype can invade. This is called an evolutionary branching point, and it implies that the phenotype will diverge in the population. Because adaptive dynamics studies invasions into monomorphic populations, we cannot use this approach to study how phenotypes evolve after they diverge. That is, we do not know how different the phenotypes will become, or how many different phenotypes might arise. However, the existence of an evolutionary branching point implies that evolution favours diversification of the focal phenotype. We now explain this in lines 130-150.

Also, I strongly feel that the reference to the development of personality does not belong to this manuscript. Social learning has very little to do with animal personality, which is the combination of many different traits, many of them being innate. The reference to the development of personality looks alien in this (otherwise pretty good) manuscript and makes me think that it was included mainly for the purpose of making the manuscript more attractive. If you need something catchy, I would rather make an emphasis on the implications to the cultural evolution instead.

We appreciate the reviewer’s point. Please see our response to the similar comment from reviewer 1.

In terms of interpretation and discussion of the results, it would be helpful to give a simple example of real-life divergence of social learning phenotypes driven by the environment alone, without pre-existing differences among individuals. The authors refer to Aplin and Morand-Ferron (2017) but do not give any examples from there, which makes it difficult to visualize it to someone who (like myself) do not have time or desire to read that additional paper.

We agree. Aplin and Morand-Ferron (2017) is an excellent paper, but we understand that most of our readers will not have time to look it up on the spot. We now explain the most relevant result of the paper the first time we mention it (lines 43-47).

And a small final remark: male sperm whales do not fail to learn coda vocalizations (as you claim on line 38). They just do not produce them, as these vocalizations are mostly used for intra-group communication, and adult males do not belong to any group.

Thank you. We have replaced the example.